# Wnt5A and TGFβ1 Converges through YAP1 Activity and Integrin Alpha v Up-Regulation Promoting Epithelial to Mesenchymal Transition in Ovarian Cancer Cells and Mesothelial Cell Activation

**DOI:** 10.3390/cells11020237

**Published:** 2022-01-11

**Authors:** Zeinab Dehghani-Ghobadi, Shahrzad Sheikh Hasani, Ehsan Arefian, Ghamartaj Hossein

**Affiliations:** 1Developmental Biology Laboratory, Department of Animal Biology, School of Biology, College of Science, University of Tehran, Tehran 1417614411, Iran; zeinabdehghani@ut.ac.ir; 2Department of Gynecology Oncology Valiasr, Imam Khomeini Hospital, Tehran University of Medical Science, Tehran 1419733141, Iran; sh-sheikhhasani@sina.tums.ac.ir; 3Molecular Virology Laboratory, Department of Microbiology, School of Biology, College of Science, University of Tehran, Tehran 1417614411, Iran; arefian@ut.ac.ir

**Keywords:** epithelial to mesenchymal transition, ovarian cancer, Wnt5A, TGFβ1/Smad signaling, hippo-YAP1/TAZ signaling, integrin alpha v, mesothelial cell activation

## Abstract

In this paper, we investigate whether Wnt5A is associated with the TGF-β1/Smad2/3 and Hippo-YAP1/TAZ-TEAD pathways, implicated in epithelial to mesenchymal transition (EMT) in epithelial ovarian cancer. We used 3D and 2D cultures of human epithelial ovarian cancer cell lines SKOV-3, OVCAR-3, CAOV-4, and different subtypes of human serous ovarian cancer compared to normal ovary specimens. Wnt5A showed a positive correlation with TAZ and TGFβ1 in high- and low-grade serous ovarian cancer specimens compared to borderline serous and normal ovaries. Silencing Wnt5A by siRNAs significantly decreased Smad2/3 activation and YAP1 expression and nuclear shuttling in ovarian cancer (OvCa) cells. Furthermore, Wnt5A was required for TGFβ1-induced cell migration and invasion. In addition, inhibition of YAP1 transcriptional activity by Verteporfin (VP) altered OvCa cell migration and invasion through decreased Wnt5A expression and inhibition of Smad2/3 activation, which was reverted in the presence of exogenous Wnt5A. We found that the activation of TGFβ1 and YAP1 nuclear shuttling was promoted by Wnt5A-induced integrin alpha v. Lastly, Wnt5A was implicated in activating human primary omental mesothelial cells and subsequent invasion of ovarian cancer cells. Together, we propose that Wnt5A could be a critical mediator of EMT-associated pathways.

## 1. Introduction

Epithelial ovarian cancer (EOC) is the most common type of ovarian cancer, in which widespread peritoneal dissemination is the main route of metastasis and ascites. The five-year survival rate is less than 35% [1]. EOC development and metastasis are associated with fibrosis [2], one of the driving forces behind epithelial-to-mesenchymal transition (EMT). Therefore, deciphering EMT regulators in EOC is urgently needed to develop new therapies to eliminate metastatic EOC and improve overall survival in these patients. 

Wnt signaling is one of the critical signaling pathways, and its deregulation is tightly associated with cancer progression [3]. The βcatenin-independent Wnt signaling known as the non-canonical pathway involves Wnt/Ca^2+^ and Wnt/planar cell polarity (PCP) pathways, which mediate cell polarity, movement, and cytoskeletal reorganization [4]. Wnt5A is mainly a non-canonical Wnt molecule that can act in various cancers as either a tumor-promoter or a tumor suppressor [5]. Our published studies and other reports showed that Wnt5A exhibits a tumor-promoting effect and could be associated with EMT in EOC progression [6,7,8]. Ovarian cancer (OvCa) metastasis requires mesothelial to mesenchymal transition (MMT) of the peritoneum. These “activated mesothelial cells” promote metastasis by supporting tumor cell adhesion, invasion, and proliferation [9]. A xenograft mouse model showed that host-derived Wnt5A promoted ovarian tumor cell adhesion to peritoneal mesothelial cells as well as migration and invasion, leading to colonization of peritoneal explants [10]. However, how Wnt5A promotes EMT in OvCa at the molecular level remains obscure. 

TGFβ plays an essential role in fibrosis and subsequent EMT through various effects involving Smad signaling [11]. The TGFβ superfamily members elicit signaling through type I and type II serine/threonine kinases receptors that form a heteromeric complex. Among the seven known mammalian type I receptors termed activin receptor-like kinase 1–7 (ALK1-7), ALK5 is expressed on many different cell types and used by TGFβ1 for signaling [12]. Extensive evidence suggests that the canonical ALK5/Smad3 pathway is critically involved in EMT and the pathogenesis of fibrosis in many tissues [13], and inhibition of ALK5 with small-molecule LY2157299 (Galunisertib) showed reduced tumor metastasis in the mouse xenograft model [14]. A previous study showed that Wnt5A is required to mediate the effects of TGFβ on the mammary gland and breast cancer [15].

Another key signaling player in fibrosis and EMT is Yes-associated protein-1 (YAP1) along with transcriptional co-activator with PDZ-binding motif (TAZ), which are inhibited by the Hippo tumor suppressor pathway [16]. Upon inhibition of the Hippo pathway, YAP1/TAZ is activated and translocated into the nucleus to bind TEAD family transcription factors to stimulate target gene expression involved in cell proliferation, stem cell self-renewal, cytoskeletal reorganization, and tumorigenesis [16]. In OvCa, elevated levels of YAP1 and TAZ have been associated with enhanced cell migration, cell proliferation, EMT, and poor prognosis [17]. Park and collaborators showed that Wnt5A is a potent activator of YAP1/TAZ, and reciprocally YAP1/TAZ-TEAD targets the Wnt5A gene [18]. To date, Wnt5A crosstalk with the TGFβ1/Smad signaling pathway and YAP1/TAZ-TEAD transcriptional activities in EOC remains unknown. 

Integrins are essential mediators of signaling crosstalk between OvCa cells and the mesothelium involved in the dissemination, invasion, and peritoneal metastasis of ovarian cancer cells [19,20]. The integrin αvβ6 was a prognostic marker in a large cohort of ovarian cancer patients [21]. Integrin αvβ6 binds to the RGD peptide present in the latency-associated peptide (LAP) associated with TGF-β1 along with latent transforming growth factor-beta binding protein 1 (LTBP1), including a conformational change of TGF-β1–LAP–LTBP1 complex known as long latency complex (LLC) [22]. Integrin αvβ6 releases TGF-β1 from the LLC, which binds to its receptor, thus activating the pathway [22]. Recently we reported that Wnt5A induced integrin αv expression in ovarian cancer cells and showed a positive relationship between Wnt5A, αv, and β6 expression in the metastatic human serous ovarian cancer specimens [23].

Additionally, inhibition of integrin αv abrogates Wnt5A-induced cell migration [23]. Our previous report was consistent with the emerging data suggesting that αvβ6 integrin regulates ovarian cancer invasion and metastasis [21,24]. The current study questioned whether Wnt5A regulates the EMT mediators: TGFβ1 and Yap1 in OvCa. We found that Wnt5A is required for TGFβ1 activation and Smad2/3 activation through enhanced YAP1 nuclear shuttling and subsequent up-regulation of integrin αv. We established that Wnt5A is an essential mediator in EMT events associated with OvCa through activation of human omentum-derived mesothelial cells, which occurs as an initial step of OvCa metastasis. 

## 2. Materials and Methods

### 2.1. Cell Culture and Reagents

SKOV-3, OVCAR-3, and CAOV- 4 cell lines (ovarian adenocarcinomas) were kindly provided by Dr. A.H. Zarnani (Avicenna Research Center, Tehran, Iran). The cells were grown in Roswell Park Memorial Institute medium (RPMI)-1640 supplemented with 10% fetal bovine serum (FBS) and 100 U/mL penicillin, and 100 μg/mL streptomycin antibiotics obtained from Life Technologies GmbH (Darmstadt, Germany) at 37 °C in 5% CO_2_ atmosphere under 90–95% humidity. Galunisertib, an inhibitor of ALK5, Verteporfin as an inhibitor of the interaction of YAP with TEAD, CWHM-12 a specific inhibitor of αv integrin, and recombinant human TGFβ1 (rhTGFβ1) were obtained from MedChemExpress (Monmouth Junction, NJ, USA). Recombinant human Wnt5A was obtained from R&D Systems (Minneapolis, MN, USA). For 3D culture, 10^5^ cells/mL were seeded in 6 well plates coated with 1% low melt agarose (IBI SCIENTIFIC, Tryon, NC, USA) in a complete medium (RPMI containing 10% FBS, 1% L-glutamine, 1% penicillin/streptomycin). The following antibodies were used in this study: mouse monoclonal anti-human pSmad2/3 and mouse monoclonal anti-human Smad2/3 were purchased from (Santa Cruz Biotechnology Inc. Heidelberg, Germany), rabbit polyclonal anti-human TGFβ1, rabbit polyclonal anti-human YAP1, and rabbit polyclonal anti-human integrin αv (Biorbyt Ltd., Cambridge, UK); mouse monoclonal anti-human Wnt5A and rabbit polyclonal anti-human GAPDH rabbit polyclonal anti-human α-SMA (Abcam, Boston, MA, USA). 

### 2.2. Stable Overexpression of Wnt5A, Transient Wnt5A Gene Knock-Down, and Real-Time qRT-PCR 

Stable cell lines that overexpress Wnt5A named as C3/OVCAR-3 and C9/SKOV-3 have been previously established [23]. Wnt5A siRNAs (ON-TARGET plus SMART pool human Wnt5A, Cat. 1349-4176 siRNA; Fisher Scientific AG, Wohlen, Switzerland) and non-target siRNAs (ON-TARGET plus SMART pool human NonTarget siRNA, Cat# 1153–7240, Fisher Scientific) were used as specific and negative (scramble) control, respectively as previously described [8,23]. Cell transfection and qRT-PCR were performed and quantified as previously described [8,23]. For the mesothelial clearance assay, a lentiviral short hairpin RNA (shRNA) targeting Wnt5a (TL320572; OriGene, Rockville, MD, USA) and a non-target shRNA as a negative control (TR30021; OriGene, Rockville, MD, USA). OVCAR-3 Cells were transduced with four different *WNT5A* human shRNA lentiviral particles or control scrambled shRNA lentiviral particles expressing GFP and puromycin (pGFP-C-shLenti), Polyethylenimine (PEI, sc-507213) was used for cell transfection. The medium was replaced after 48 h with a medium containing 1.2 μg/mL puromycin for seven days. The GFP expressed in the lentiviral vectors was used to quantify transduction efficiency and, qRT-PCR and western blotting were used as previously described [23]. To select the transduced cells with the highest knock-down efficiency. The primer sequences are listed in Appendix A.

### 2.3. Treatments

For recombinant Wnt5A treatment, 600 ng/mL of rhWnt5A was added overnight. For TGFBRI inhibition, cells were pre-treated with 10 µM of Galunisertib for 48 h before additional treatment. Verteporfin (VP) was used as an inhibitor of YAP1-TEAD interaction to repress YAP1′s function, and cells were pre-treated with 5 µM VP for 1–2 h before additional treatment. Treatment with rhTGFβ1 (10 ng/mL) was performed for 1 h to assess the Smad2/3 activation or YAP1 cellular localization in control (without pre-treatment) or pre-treated cells with VP or Galunisertib. For integrin αv inhibition, cells were treated with CWHM-12 (10 μM) for 24 h. 

### 2.4. Immunofluorescence and Western Blot Analysis

To assess pSmad2/3 or YAP1 subcellular localization SiRNA Wnt5A or siRNA scrambled transfected and non-transfected cells were treated with Galunisertib alone or in combination for 48 h. Cells were then fixed and blocked as previously described [23] and incubated with anti-pSmad2/3 antibody (1:100) or anti-YAP1 antibody (1:100) overnight at 4 °C, and immunostaining was performed as previously described (23). For the Western blot, cells were lysed, and protein content was measured as previously described [23]. It was performed with anti-TGFβ1 (1:1000), anti-Wnt5A (1:1000), anti-pSmad2/3 (1:1000), anti-YAP1 (1:2000), anti-α-SMA (1:1000), anti-Integrin αv (1:1500), and anti-GAPDH (1:1000) antibodies for overnight at 4 °C as previously described [23].

### 2.5. Scratch Assay and Transwell Migration/Invasion Assays

Scratch assay, transwell cell migration, and invasion assays were performed using siRNA against Wnt5A transfected cells compared to scrambled siRNA in the presence or absence of rhTGFβ1 or VP for the indicated times and quantified as previously described [23].

### 2.6. Tumor Specimens

Human ovarian specimens were collected from Imam Khomeini Hospital Complex, Tehran, Iran. The characteristics of the patients are described in Appendix A. A total of 41 specimens comprised six normal (N), 10 borderline-serous ovarian cancers (BLSOCs), 10 low-grade serous ovarian cancers (LGSOCs), and 15 high-grade serous ovarian cancers (HGSOCs). These samples were used to assess the mRNA levels of WNT5A, ROR1, ROR2, TGFB1, TGFB2, TGFBR1, TGFBR2, YAP1, TAZ, CCN1 and CCN2 using real-time qRT-PCR. The list of primer sequences is displayed in Appendix A. 

### 2.7. Human Primary Omental Mesothelial Cells Isolation, Mesothelial Cell Activation, and Clearance Test

After receiving informed consent, small pieces of human omentum were removed from patients undergoing abdominal surgery for noninfectious and non-cancerous conditions at Imam Khomeini University Hospital Complex. Immediately after removal, the pieces of omentum were placed in prewarmed medium MCDB105/M199 (Sigma), washed in phosphate-buffered saline, then cut into small pieces (1 mm^3^) and treated with 0.05% trypsin-0.02% EDTA (CWHM-12) for 30 min at 37 °C under gentle shaking. The detached mesothelial cells were pelleted by centrifugation at 300 G for 5 min, resuspended in MCDB105/M199 supplemented with 10% fetal bovine serum (FBS) (GIBCO, Amarillo, TX, USA), and amphotericin B (2.5 μg/mL) (Bristol-Myers Squibb, NY, USA) and seeded in 1% gelatin-coated plate. After 14–21 days, human primary omental mesothelial cells (HPOMC) formed confluent monolayers, which were routinely passaged at a 1:3 split ratio using 0.05% trypsin–0.02% EDTA. HPOMCS were characterized using qRT-PCR for the expression following mesothelial markers: CALB2 (Calretinin), KRT 18 Cytokeratin18, CDH1 (E-cadherin), MSLN (Mesothelin), CDH2 (N-cadherin), FN1 (Fibronectin1), VIM (Vimentin), and lack of expression of FAP (Fibroblast activation protein). The primer sequences used for qRT-PCR were displayed in Appendix A. 

HPOMCS was used at low passage (P) (1–3) and seeded for the mesothelial clearance test. OVCAR-3 cells (10^5^ cells) were resuspended in RPMI containing CellTracker™ CM-DiI at 1 µg/mL (Thermo Fisher Scientific, Waltham, MA, USA), incubated for 45 min at 37 °C and then 15 min at 4 °C, and washed two times with PBS. The stained cells were suspended in RPMI with 10% FBS, and 20% methylcellulose for 3D culture using hanging drop (1000 cells/drop) as previously described [25]. After 48 h, the OVCAR-3 spheroids (3–5/ well) were collected and gently added on top of the HPOMCS monolayer. OVCAR-3 spheroids disintegrated, and invaded cells formed holes in the HPOMCS monolayer. Disaggregation of spheroids was quantitated by manually outlining the area (μm) of each spheroid at 2 h set as the initial time point and after 24, 48 and, 72 h using ImageJ and, calculated as follows: the area of spheroids after 24, 48 or 72 h divided by spheroid’s area at 2 h multiplied by 100. The results are expressed as mean +/− SD from at least three independent experiments.

### 2.8. GO and KEGG Pathway Enrichment Analysis

Gene Ontology (GO) analysis and Kyoto Encyclopedia of Genes and Genomes (KEGG) analysis were performed for TGFβ and Wnt5A pathways using Enrichr (http://amp.pharm.mssm.edu/Enrichr, accessed on 5 September 2021), an intuitive enrichment analysis web-based tool that includes curated gene-set libraries, a computational method for inferring knowledge about an input gene set by comparing it to annotated gene sets representing prior biological knowledge [26,27,28]. We applied Enrichr to enrichment analysis of transcription factors (TFs) and GO in up-regulated genes, including the TGFβ family, Wnt5A, and YAP/TAZ components in human ovarian cancer samples. A *p*-value of <0.05 was considered statistically significant. 

### 2.9. Hierarchical Clustering Analysis and PPI Network Construction

A bidirectional hierarchical clustering heatmap of assessed genes in human serous ovarian cancer specimens was constructed using the gplots package of R language after extracting the expression values from the gene expression profile. Terms with *p* < 0.05 were collected and grouped into clusters based on their membership similarities. More specifically, *p*-values were calculated based on the cumulative hypergeometric distribution. The most significant term within a cluster was selected as the one representing the cluster. Subsequently, a functional enrichment analysis was performed . Biological process (BP), and KEGG pathway enrichment was performed to determine the involvement of genes in different biological pathways by using Enrichr GO terms. Annotated results for GO (BP) are displayed in a table. Construction of the Protein–Protein Interaction (PPI) network STRING (Search Tool for the Retrieval of Interacting Genes/Proteins, v11.0, http://www.string-db.org, accessed on 11 October 2021) database composed of experimental data, computational prediction method with a confidence score > 0.4 (medium confidence), was used to retrieve interacting genes.

### 2.10. Statistical Analysis

The normality of nominal variables was analyzed by performing the Kolmogorov–Smirnov test. Skewed and normal distributed metric variables were analyzed using Mann–Whitney U or among multiple groups using Kruskal–Wallis and one-way ANOVA tests, respectively, using R version 3.5.2. The Pearson correlation coefficient test analyzed correlations between gene expressions. All experiments were performed at least three times, and the results were expressed as mean +/− SD; *p* < 0.05 was considered significant. 

## 3. Results

### 3.1. Higher Expression Levels of Different Components of TGFβfamily, Wnt5A, and Hippo-Related Genes in HGSOC Specimens

We performed the qRT-PCR analysis of TGFβ components (TGFB1, TGFB2, TGFBR1, TGFBR2), Wnt5A/ROR1/ROR2, and Hippo-related genes (YAP1, TAZ, CCN1, and CCN2) in serous ovarian cancer subtypes (LGSOC, HGSOC, BLSOC) compared to the normal ovary. Figure 1A shows a heatmap plot to classify the up-regulated and down-regulated analyzed genes for normal and different serous ovarian cancer subtypes. Next, a two-factor experimental design was developed to evaluate the gene type expression difference in LGSOC, HGSOC, BLSOC, and normal groups. We found that the expression levels of WNT5A, ROR1, ROR2, TGFB1, TGFB2, TGFBR1, TGFBR2, YAP1, TAZ, CCN1, and CCN2 were significantly different between normal and cancerous groups *(p ˂* 2^−16^*)*. HGSOC group showed a higher median expression level of assessed genes than other groups (Figure 1B). These results have prompted us to look for the existence of a relationship between YAP1/TAZ, Wnt5A, and TGFβ components in the serous ovarian cancer subtypes with the ability to metastasize (LGSOC + HGSOC). The correlation analysis between these genes showed that YAP1 positively correlated with TGFBR1 (r = 0.41, *p* = 0.041), ROR2 (r = 0.52, *p* = 0.007) and ROR1 (r = 0.48, *p* = 0.016) (Figure 1C). Wnt5A positively correlated with TAZ (r = 0.39, *p* = 0.016) and TGFβ1 (r = 0.45, *p* = 0.022 (Figure 1C) while TAZ positively correlated with TGFBR2 (r = 0.43, *p* = 0.031) (Figure 1C). The top 20 enriched common GO terms (BP) between Wnt5A, TGFβ1, TGFβ2, TGFBR1, and TGFBR2 ranked by fold enrichment, and enrichment score is listed in Appendix A. It revealed that the most important module was mainly associated with the regulation of the EMT process. Furthermore, Protein–Protein Interaction (PPI) network STRING analysis showed a strong interaction among TGFB1, TGFBR1, TGFBR2, and YAP1 (Figure 1D). It is worth noting that the KEGG database revealed that, among the five common pathways between Wnt5A and the TGFβ family, the Hippo signaling pathway was one of the regulators among the common pathway (hsa04390, *p*-value: 2.454^−11^) (Figure 1E). 

### 3.2. Wnt5A Modulates Smad2/3 Activation in OvCa Cells

First, we assessed the expression levels of TGFβ1, TGFβ2, TGFBR1, TGFBR2 Wnt5A, YAP1, and TAZ in SKOV-3, OVCAR-3, and CAOV-4 cell lines in multicellular aggregates (MCAs) culture conditions. The OvCa cells were transfected with siRNA against Wnt5A, leading to a significant reduction of the levels of TGFβ1, and pSmad2/3 compared to scramble (Scr) in 3D culture condition (MCAs) (Figure 2A, left and right panels). Correspondingly, immunostaining of pSmad2/3 nuclear localization was significantly reduced in Wnt5A silenced cells compared to Scr (Figure 2B, left and right panels). Treatment of cells with rhTGFβ1 showed increased pSmad2/3 immunostaining (Figure 2B, left and right panels), and treatment of Wnt5A silenced cells with rhTGFβ1 showed a weak pSmad2/3 nuclear immunostaining compared to control cells treated with rhTGFβalone (Figure 2B, left and right panels).

### 3.3. Wnt5A Is Required for TGFβ1-Induced Migration and Invasion of OvCa Cells

Next, we sought to determine whether Wnt5A could mediate TGFβ1-induced cell migration and invasion. We observed significantly increased cell migration and invasion in OvCa cells treated with rhWnt5A or rhTGFβ1 (Figure 3A–C). A significantly decreased cell migration and invasion were observed in Wnt5A silenced cells which could not be reverted upon adding exogenous TGFβ1 compared to control (Figure 3A–C). Cell migration and invasion are related to EMT events; thus, we assessed the EMT markers regulated directly by TGFβ Smad2, 3 such as N-cadherin, Vimentin, fibronectin, and CD44 as reported by Enrich r tools (http://amp.pharm.mssm.edu/Enrichr, accessed on 11 October 2021). Snail and slug are well-known repressors of E-cadherin expression and could be regulated indirectly by the TGFβ signaling pathway [29]. In SKOV-3 cells, CDH2 (N-cadherin), VIM (Vimentin), FN (Fibronectin), and CD44 increased 2.2-fold, 5.3-fold, 2.5-fold, and 1.7-fold in the presence of rhTGFβ1 compared to control, respectively (Appendix A, upper panel). SNAI2 (Slug) and CDH2 expression levels significantly increased by 4.5-fold and 8.6-fold in rhTGFβ1-treated OVCAR3 cells compared to control, respectively (Appendix A, middle panel). We found a significant increase of SNAI2, FN, and VM expression levels: 2.2-fold, 2.8-fold, and 1.8-fold in rhTGFβ1-treated CAOV-4 cells compared to control, respectively (Appendix A, lower panel). In contrast, treatment of cells with Galunisertib, an ALK5 inhibitor, decreased expression levels of most EMT markers (Appendix A). Inhibition of endogenous TGFβ1 by pre-treatment of cells with Galunisertib before adding rhTGFβ1 led to potent induction of EMT markers in OvCa cells (Appendix A). Moreover, decreased expression of most EMT markers in Wnt5A silenced cells were partially reverted in the presence of rhTGFβ1 (Appendix A).

### 3.4. Positive Feedback Loop between Wnt5A and YAP1 and Inhibition of YAP1 Transcriptional Activity Decreases Smad2/3 Activation in OvCa Cells

Park and collaborators previously showed that Wnt5A activates YAP1/TAZ and is a downstream target gene of YAP1/TAZ–TEAD, indicating a potential positive feedback loop [18]. To date, evidence for a link between Wnt5A and YAP1/TAZ in EOC is missing. YAP1 expression levels were decreased by 30, 83, and 70% in Wnt5A silenced MACs of SKOV-3, OVCAR-3, and CAOV-4 cells compared to control, respectively (Figure 4A, upper, and lower panels). Similarly, there were significantly decreased YAP1, TAZ, CCN1, and CCN2 mRNA levels in Wnt5A silenced MCAs of OvCa cells compared to control (Appendix A). To assess the impact of YAP1 transcriptional activity on Wnt5A expression levels in OvCa cells, we used Verteporfin (VP) as an inhibitor of YAP1/TAZ–TEAD interaction. We found that VP down-regulates Wnt5A at protein and mRNA levels (Figure 4B left and right panels and Appendix A) along with the downstream target genes CCN1 and CCN2 in MCAs of OvCa cells (Appendix A). Together, these observations suggest that Wnt5A could regulate YAP1 expression levels and is also a downstream target of YAP1. Next, we questioned whether YAP1 transcriptional activity is required for direct Smad2/3 activation or indirectly through Wnt5A. OvCa cells were treated with exogenous TGFβ1 or Wnt5A alone or with VP, then pSmad2/3 and YAP1 levels were assessed in MCAs of OvCa cells. We found a significant and robust inhibition of Smad2/3 and YAP1 activation in VP-treated cells in OvCa MCAs (Figure 4C, upper and lower panels). In addition, rhWnt5A or rhTGFβ1 treatment of VP-treated cells could partially revert Smad2/3 activation in OvCa cells except in OVCAR-3 cells (Figure 4C upper and lower panels). Although rhWnt5A or rhTGFβ1 increased YAP1 expression, rhWnt5A could potently revert the down-regulated YAP1 expression in VP-treated cells even more (Figure 4C upper and lower panels). 

Similarly, we found significantly increased pSmad2/3 and YAP1 nuclear immunostaining in rhWnt5A-treated cells and a significantly decreased nuclear immunostaining of pSmad2/3 and YAP1 in VP-treated cells (Figure 5A–C left and right panels), but it was partially reversed in the presence of rhWnt5A (Figure 5A–C left and right panels). These results suggest that YAP1 could mediate Wnt5A-induced TGFβ1/Smad2/3 activation.

A significantly decreased cell migration of OvCa cells was observed in VP-treated cells reverted upon adding rhWnt5A (Appendix A). However, cell invasion was significantly decreased in SKOV-3 VP-treated cells, and no significant effect was observed in VP-treated OVCAR-3 and CAOV-4 cells (Appendix A, upper and lower panels). Furthermore, EMT markers: CDH2, SNAI1, SNAI2, FN, VIM, and CD44 were decreased at various levels in VP-treated OvCa cells and reversed by the presence of rhWnt5A (Appendix A). These results suggest that Wnt5A is critical for YAP1/TAZ–TEAD- induced EMT in OvCa cells. 

### 3.5. YAP1 Regulates Wnt5A-Induced Integrin av and Smad2/3 Activation

Our previous study showed up-regulation of integrin αv expression in Wnt5A overexpressing OvCa cells associated with cell proliferation, migration, and fibronectin attachment [23]. It is well known that αv integrins, particularly αvβ6 and αvβ8, are specialized activators of TGFβ [30,31]. Here we questioned whether Wnt5A-induced integrin αv contributed to TGFβ activation in OvCa cells. 

As the first step, we found that Wnt5A knock-down significantly decreased ITGAV, ITGB8 in MCAs culture condition, and ITB6 expression levels in MCAs Wnt5A-silenced cells compared to scramble (Appendix A). Similarly, integrin αv levels were significantly decreased in MCAs of Wnt5A-silenced cells (Figure 6A, upper and lower panels). Next, we questioned whether inhibition of YAP1 transcriptional activity might influence integrin αv expression. There were significantly increased expression levels of integrin αv in rhWnt5A-treated MCAs OvCa cells (Figure 6B, left and right panels). Moreover, we noticed significantly decreased integrin αv in VP-treated cells and interestingly reverted in the presence of exogenous Wnt5A (Figure 6B, left and right panels). We previously established a stable overexpressed Wnt5A clone of OVCAR-3 cells (C3/OVCAR-3) and SKOV-3 cells (C9/SKOV-3) [23]. We asked whether integrin αv is associated with Smad2/3 and YAP1 activation by Wnt5A. To answer this question, we used CWHM-12, a pan inhibitor of integrin αv. It was remarkable that the inhibition of integrin αv in Wnt5A overexpressing cells led to a significant decrease in Smad2/3 activation as well as YAP1 and integrin αv expression levels compared to untreated cells in MCAs OvCa cells (Figure 6C, left and right panels). It is of particular interest that TGFβ1 expression levels remain unchanged in the CWHM-12-treated cells, indicating that integrin αv impacts TGFβ1 activation (Figure 6C, left and right panels). These results were confirmed by the reduced nuclear staining of pSmad2/3 and YAP1 in CWHM-12-treated cells compared to untreated cells (Figure 6D, left and right panels). Also, TAZ, CCN1, and CCN2 mRNA levels were significantly reduced in CWHM-12-treated Wnt5A overexpressed clones (Appendix A, left and right panels). Correspondingly, EMT markers such as SNAI1, FN, VIM, and CD44 were significantly decreased in CWHM-12-treated clones compared to untreated control cells (Appendix A, left and right panels). Moreover, cell migration and invasion were significantly reduced by 50% in Wnt5A overexpressed clones in the presence of CWHM-12 compared to control (Appendix A, left and right panels). 

### 3.6. Wnt5A Induce Mesothelial Cell Activation and Clearance through Smad2/3 and YAP1 Activation

Ovarian cancer cells use the contractility of integrin and actinomycin to exert force on fibronectin in the mesothelial monolayer, achieving gaps between mesothelial cells by retracting these cells, a phenomenon known as this mesothelial clearance [31]. OvCa cell lines have been profiled as mesothelial clearance–competent and –incompetent based on the enrichment of mesenchymal genes, including SNAI1, 37ZEB1, and TWIST1 in the clearance-competent cells [32]. However, the involvement of Wnt5A in mesothelial activation and clearance remains unknown. We ascertained that the isolated HPOMCS express specific mesothelial markers: CALB2 (Calretinin), Cytokeratin18, CDH1 (E-cadherin), MSLN (Mesothelin), CDH2 (N-cadherin), FN1 (Fibronectin1), VIM (Vimentin), and lack of expression of FAP (Fibroblast activation protein) (Appendix A). Moreover, HPOMCSs showed spindle-shaped morphology (Appendix A) and expressed higher αSMA levels in the presence of rhWnt5A or conditioned medium (CM) isolated from C3/OVCAR-3 (Figure 7A, left and right panels). Similarly, rhWnt5A or conditioned medium (CM) isolated from C3/OVCAR-3 could activate Smad2/3 and YAP1 activation in HPOMCS as revealed by immunofluorescence (Figure 7B,C, left and right panels). These changes were not observed when HPOMCSs were treated with CM isolated from Wnt5A-silenced OVCAR-3 cells (Figure 7A–C). Correspondingly higher expression levels of EMT markers: CDH-2, FAP, and ACTA2 were up-regulated in HPOMCS treated with rhWnt5A or CM from C3/OVCAR-3 (Appendix A). 

The mesothelial clearance was investigated using C3/OVCAR-3 spheroids seeded on the top of the HPOMCSs layer to assess disaggregation and invasion of cancer cells through it for 24, 48, and 72 h compared to the initial time set as two hours. After 48 and 72 h, we found a significantly increased surface area indicating disaggregation of C3/OVCAR-3 spheroids and invasion through activated mesothelial cells compared to parental OVCAR-3 cells (Figure 8A,B). This disaggregation was inhibited in either VP- or CWHM-12-treated OVCAR-3 cells (Figure 8A,B). These findings may establish that Wnt5A-induced integrin αv mediates mesothelial cell activation and clearance through YAP1 activation. 

## 4. Discussion

In this study, we unveiled for the first time the molecular mechanism of Wnt5A-induced EMT in OvCa. We were able to identify a functional connection between Wnt5A and TGFβ1 converging through YAP1 activities and integrin αv up-regulation. We reported that Wnt5A induced YAP1 activity, further required for Smad2/3 activation. Inhibition of YAP1/TAZ-TEAD interaction downregulated Wnt5A expression, indicating that Wnt5A is a target gene of YAP1/TAZ-TEAD in a positive feedback loop for OvCa cells. Furthermore, Wnt5A up-regulated integrin αv, β6, and β8 expression, activating TGFβ1. Wnt5A mediated TGFβ1-induced EMT in OvCa cells and mesothelial cell clearance via two mechanisms: (1) through YAP1 up-regulation and (2) by up-regulation of integrin αv implicated in TGFβ1 activation. 

Chronic inflammation is associated with fibrosis, a significant risk factor in EOC incidence and progression [33]. TGFβ1 is one of the primary signaling cascades implicated in fibrosis often associated with chronic inflammation [34]. Wnt5A may promote inflammation and fibrosis in multiple organs and tissues [35,36]. Our previous work showed that Wnt5A promotes EMT and contributes to a pro-inflammatory microenvironment promoting OvCa cell migration [37]. Here, the bioinformatics analysis showed that both Wnt5A and TGFβ signaling pathways could be regulated by the Hippo signaling pathway (Figure 1E), and a similar result was obtained for Wnt5A/ROR1/ROR2 and TGFβ pathway (data not shown). Concomitantly, HGSOC specimens showed higher expression levels of ROR1, ROR2, Wnt5A, TGFβ1, TGFβ2, TGFBR1, TGFBR2, YAP1, and TAZ in HGSOC specimens compared to other serous histological subtypes and normal ovaries (Figure 1B). 

Correspondingly, it has been demonstrated that TGFβ1 induces EMT in OvCa cells [37]. and contributes to the progression of borderline serous tumors to low-grade tumors. Direct regulation of Wnt5A by TGFβ1 and YAP1/TAZ-Tead was verified in primary cells in culture as Smad, and TEAD binding sites were identified in the Wnt5A promoter [38,39]. Evidence for Wnt5A crosstalk with TGFβ1 was shown in mammary gland development and breast cancer, given that Wnt5A was required for TGF-β acting through antagonizing β-catenin, thereby limiting mammary gland morphogenesis [15]. This restrained stem or progenitor cell population provides a novel mediator for TGFβ tumor-suppressive effects [40]. It is essential to note that TGFβ1 and Wnt5A are complex and depend on microenvironmental factors. 

This study is the first evidence to show that Wnt5A is required for TGFβ1-induced EMT in ovarian cancer. A key finding made here is that down-regulation of Wnt5A inhibits Smad2/3 activation and TGFβ1 function (Figure 2). Wnt5A requirement for TGFβ1 functional activity was further validated by abrogating TGFβ1-induced EMT markers and subsequent cell migration and invasion (Figure 3). These results are further reinforced by a positive correlation between Wnt5A and TGFβ1 in HGSOC specimens (Figure 1C). 

YAP1 and TAZ constitute another family of transcriptional regulators controlled through inhibitory phosphorylation by the LATS1/2 kinases in the Hippo pathway [16]. It has been demonstrated that TAZ and YAP1 are associated with OvCa progression and poor prognosis [16,41]. Likewise, analysis of the multidimensional genomics data indicated that YAP1 and TEAD genes were frequently amplified and up-regulated in ovarian HGSOC, and Hippo/YAP1 signaling pathway played a critical role in the initiation and progression of fallopian tube-derived ovarian HGSOC [41]. Here we found that Wnt5A and YAP1 regulated each other in a positive feedback loop (Figure 4 and Figure 5), and the inhibition of YAP1 transcriptional activity by Verteporfin down-regulated Wnt5A expression and Smad2/3 activation (Figure 4 and Figure 5). In line with our findings, a recent in silico analysis revealed two regions with potential binding sites for YAP1/TEAD1 and YAP1/TEAD4, upstream of the WNT5A gene [39]. Consistently, a remarkable study by Park et al. showed that Wnt5A is not only an upstream activator but also a downstream target gene of YAP1/TAZ-TEAD, indicating a potential positive feedback loop among these genes [18]. Interestingly, Wnt5A can activate YAP1/TAZ via ROR1-FZD2/5 co-receptors signaling and at the molecular level G protein-coupled receptors Gα12/13 and their downstream Rho GTPases effectors directly connect Wnt5A to YAP/TAZ, inhibiting Lats1/2 and thereby, YAP/TAZ nuclear translocation [18]. In another study, YAP1 was found to regulate ROR1 expression, and Wnt5A/ROR1–YAP1/TAZ feedback loop was associated with cancer stem cell phenotype, tumor progression, metastasis, and, ultimately, drug resistance of breast cancer [42]. It should be noted that YAP and TAZ lack a nuclear localization signal (NLS) [16], and the machinery for their nucleocytoplasmic shuttling and nuclear accumulation is unknown. It has been suggested that any phosphorylation-independent YAP/TAZ inhibitory event may leave YAP/TAZ available for phosphorylation by LATS [16]. Thus, in our model, it is tempting to propose that Wnt5A may inhibit LATS1/2 leading to YAP1 nuclear shuttling and transcriptional activity. 

Of particular interest, we noticed a positive correlation between YAP1 and ROR1/ROR2 along with TAZ and Wnt5A expression in HGSOC specimens (Figure 1C). These findings may further reinforce the existence of a possible Wnt5A-RORs-YAP1/TAZ loop promoting ovarian cancer metastasis. Further elucidation of the Wnt5A-ROR1-YAP1/TAZ-TGFβ1 signaling axis will expand mechanistic insights and provide therapeutic targets in EOC metastasis. 

A very recent study proved that YAP1/TAZ alters Smad nuclear accumulation by acting directly or indirectly as a retention factor and alters TGFβ receptor activity through a post-translational mechanism [43]. However, the inverse is not valid as YAP1/TAZ levels in the cytoplasm or nucleus remain constant regardless of the dose of TGFβ or treatment time as long as Hippo pathway activity was constant [43]. TAZ could bind Smad2/3 through the coiled–coil region, and this interaction may dictate the subcellular localization of Smad2/3 [16,44,45]. Here, we could not exclude TGFβ pathway regulation by YAP1/TAZ–Tead since the Genes–TFs regulatory network-based analysis revealed that TGFβ2, TGFBR2, SMAD3, and SMAD7 harbored two TEAD-binding sites (data not shown). 

We can conclude that Wnt5A/YAP1 positive feedback loop mediates TGFβ1-induced EMT based on the following reasons. First, Wnt5A up-regulates YAP/TAZ expression in OvCa cells and vice versa. Second, disruption of YAP/TAZ–TEAD interaction could markedly inhibit Smad2/3 activation, which could be partially reversed with exogenous Wnt5A in VP-treated cells. Third, Wnt5A knock-down reduces Smad2/3 activation, which the presence of exogenous TGFβ1 could not rescue. Fourth exogenous TGFβ1 could not revert inhibition of cell migration and invasion in Wnt5A silenced cells, while inhibition of cell migration and invasion in VP-treated cells could be reversed upon rhWnt5A addition.

We recently showed that Wnt5A up-regulates integrin αv expression and activation, promoting OvCa cell proliferation, migration, and fibronectin attachment in 2D and 3D culture models [23]. The αv-integrins are one of the primary activators of latent-TGFβ, most prominently αvβ6 and αvβ8 [46]. Integrin αvβ6 is up-regulated in ovarian cancer [21] and is a driver of EMT, which activates TGF-β1 through binding to the RGD motif contained within LAP, which is linked to inactive TGF-β1 [47]. 

Thus, we hypothesized that Wnt5A-induced integrin αv could mediate TGFβ1 activation, supported by the fact that inhibition of integrin αv down-regulates YAP1 and Smad2/3 activation with no effect on TGFβ1 expression levels (Figure 6). Reciprocally, inhibition of YAP1 transcriptional activity largely down-regulated integrin αv (Figure 6). Interestingly, αvβ3 was highly overexpressed in the course of tumor progression and regulated cell/ECM stiffness, tumor progression, and cell/ECM stiffness and cell contractility [48]. Indeed, YAP/TAZ responded to changes in ECM stiffness: a rigid ECM translocated YAP/TAZ to the nucleus and kept them active, while the more compliant matrices favor YAP/TAZ inactivation [16]. This relation of stiffness and YAP1 subcellular localization may explain how the inhibition of integrin αv led to decreased YAP1 expression and activity levels in our model.

The peritoneal cavity and the omentum are lined by the mesothelium, a monolayer of mesothelial cells resting on a basement membrane. Mesothelial cell invasion is a rate-limiting step in OvCa early metastasis events [49].

In coculture, OVCA spheroids use myosin-generated force to retract and clear mesothelial cells, thereby exposing the underlying ECM, particularly fibronectin, for OvCa cells attachment [33,34,35,36,37,38,39,40,41,42,43,44,45,46,47,48,49,50]. Peritoneal mesothelial cells and visceral adipose secrete Wnt5A, and conditional knockout of WNT5A in mesothelial cells, significantly reduced the peritoneal metastatic tumor burden [10]. We showed the first evidence that Wnt5A induced Smad2/3 activation, YAP1 activation, αSMA, and a spindle-like shape in HPOMCS (Figure 7). These were related to MMT events in mesothelial cell activation and clearance. This report showed that Wnt5A overexpression enhances a mesothelial clearance–competent profile to the OvCa spheroids through YAP1 activation and integrin αv induction (Figure 8).

## 5. Conclusions

Wnt5A may be a critical, functional predeterminant of EMT, supporting mesothelial cell activation and retraction, leading establishing the first metastatic colony on the omentum/peritoneum. Based on our combined data, as depicted in Figure 9, we proposed a model in which OvCa cell-derived Wnt5A may inhibit YAP1 phosphorylation, which translocates into the nucleus, induces TGFβ1 and integrin αv, and Wnt5A into a positive feedback loop. Integrin αv, in turn, activates the extracellular latent-TGFβ1 as a central player in the EMT process. Moreover, YAP1 transcriptional activity required for Smad2/3 activity may come through retention of pSmad2/3 in the nucleus.

The task at hand lies in identifying key receptors that interact with Wnt5A and their downstream mediators contributing to TGFβ1/Smad and YAP1/TAZ activities in OvCa cells. The concept presented here should be relevant when selecting clinically beneficial targeted therapies.

## Figures and Tables

**Figure 1 cells-11-00237-f001:**
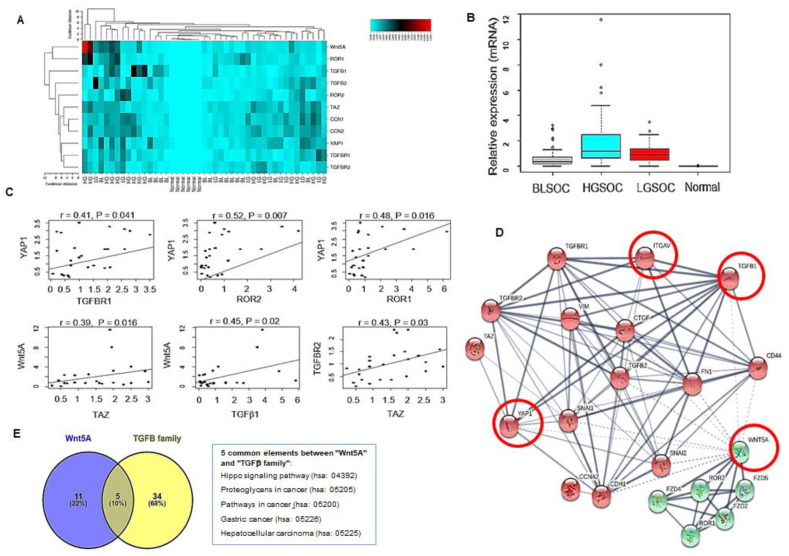
Higher expression levels of TGFβ components, Wnt5A, and Hippo-related genes in HGSOC specimens. (**A**) The heatmap shows the expression levels of TGFβ components, Wnt5A, and Hippo-related genes in Serous human EOC specimens (HG: high grade; LG: low grade; BL: borderline) and normal ovaries obtained by hierarchical cluster analysis. Each column in the figure represents a sample, and each row represents a gene. The colors in the graph indicate the magnitude of gene expression in the sample. The black–red gradient indicates that the genes are medium-high, and the blue indicates low gene expression. (**B**) The box plot shows differential gene expression levels in tumor specimens compared to normal ovaries. (**C**) The correlation analysis between TGFβ components, Wnt5A, and Hippo-related genes. (**D**) The protein–protein interaction (PPI) network of Wnt5A and TGFβ signaling components showed the interaction between the molecules involved in these signaling pathways; the thickness of the edge indicates a strong interaction between the two proteins. PPI enrichement *p* value = 1^−16^. (**E**) The Venn diagram shows the common pathways between TGFβ and Wnt5A signaling.3.2. Wnt5A Modulates Smad2/3 Activation in OvCa Cells.

**Figure 2 cells-11-00237-f002:**
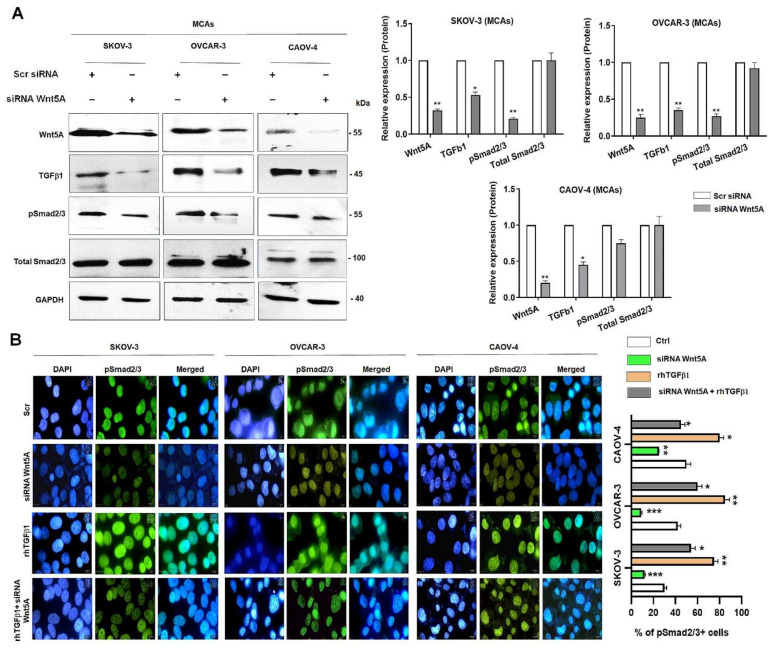
Wnt5A modulates Smad2/3 activation in ovarian cancer cells. The SKOV-3, OVCAR-3, and CAOV-4 cells were transfected with siRNA Wnt5A or Scrambled siRNA (Scr). (**A**) The expression level of Wnt5A, TGFβ1, pSmad2/3 was assessed by immunoblotting in multicellular aggregates (MCAs) (Left panel). GAPDH levels were used as an internal control. The right panel shows the quantification of bands from three independent experiments (**B**) Immunolocalization of pSmad2/3 in Wnt5A silenced cells in the absence or presence of rhTGFβ1 (10 ng/mL) for 1 h (left panel). The right panel shows the percent of pSmad2/3 positive cells. Original magnification, ×400. *: *p* < 0.05, **: *p* < 0.01 and ***: *p* < 0.001 compared to scramble (Scr), n = 3.

**Figure 3 cells-11-00237-f003:**
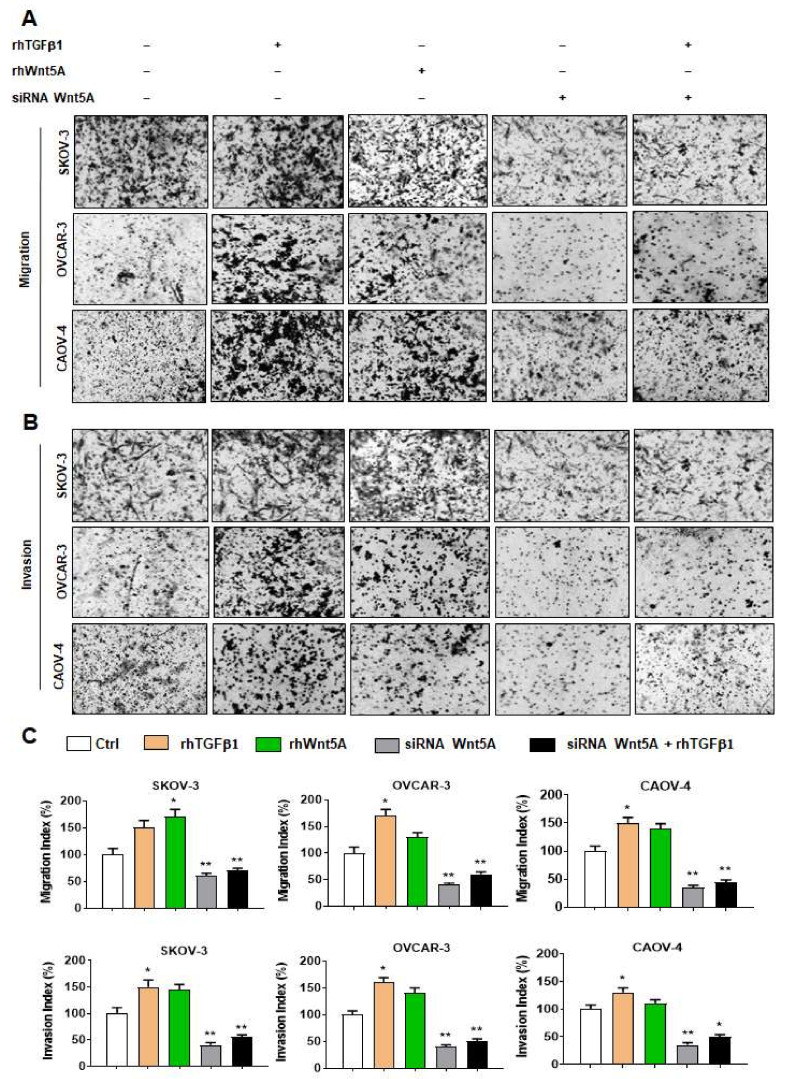
Wnt5A is required for TGFβ1-induced migration and invasion of ovarian cancer cells. (**A**) Cell migration and (**B**) Cell invasion siRNA Wnt5A or siRNA scrambled transfected cells (Scr) were seeded at 2.5 × 10^4^ cell density on the upper chamber of transwells in the presence or absence of rhTGFβ1 (10 ng/mL) or rhWnt5A (600 ng/mL) in serum-free medium for 14 h. Cell invasion was performed using matrigel-coated transwells. Photos represent one of the three independent experiments. (**C**) Migrated and invaded cells were quantified, and the results were expressed as mean ± SD, n = 3. Original magnification,×100. *: *p* < 0.05, **: *p* < 0.01 compared to untreated control cells (Ctrl) or Scr.

**Figure 4 cells-11-00237-f004:**
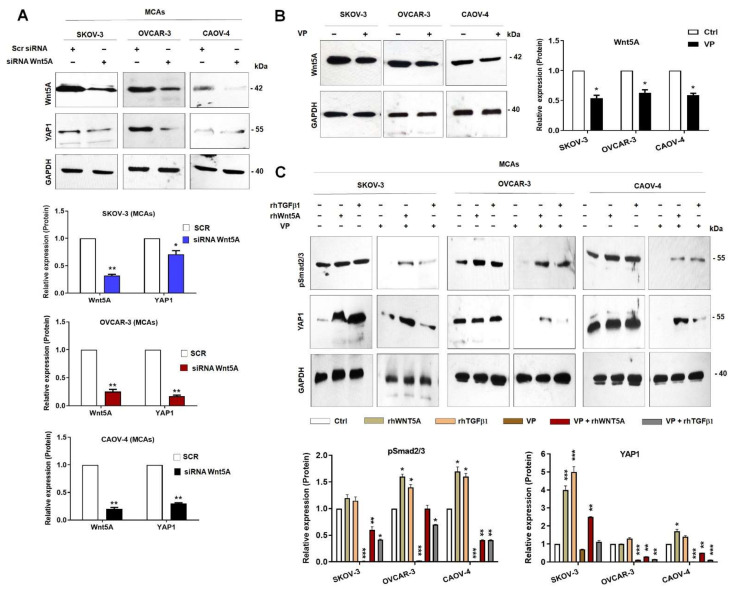
The positive feedback loop between Wnt5A and YAP1 and inhibition of YAP1 transcriptional activity decreases Smad2/3 activation in OvCa cells. SKOV-3, OVCAR-3, and CAOV-4 cells were transfected with siRNA scrambled (Scr) or siRNA Wnt5A. (**A**) The upper panel shows YAP1 expression levels in multicellular aggregates (MCAs) of OvCa cells. The lower panels show the quantification of bands from three independent experiments. (**B**) The left panel shows the expression level of Wnt5A in the cells treated with VP (5 μM), and the right panel shows the quantification of bands from three independent experiments. (**C**) Cells as MCAs were treated as follows: rhTGFβ1 (10 ng/mL) for 1 h, rhWnt5A (600 ng/mL) for 14 h, VP (5 μM) for 1 h, VP-pretreated + rhWnt5A, and VP-pretreated + rhTGFβ1. The upper panel represents immunoblots, and the lower panels show the quantification of bands from three independent experiments. GAPDH levels were used as internal control, and results are expressed as mean ± SD. *: *p* < 0.05, **: *p* < 0.01 and ***: *p* < 0.001 compared to untreated control cells (Ctrl) or Scr, n = 3.

**Figure 5 cells-11-00237-f005:**
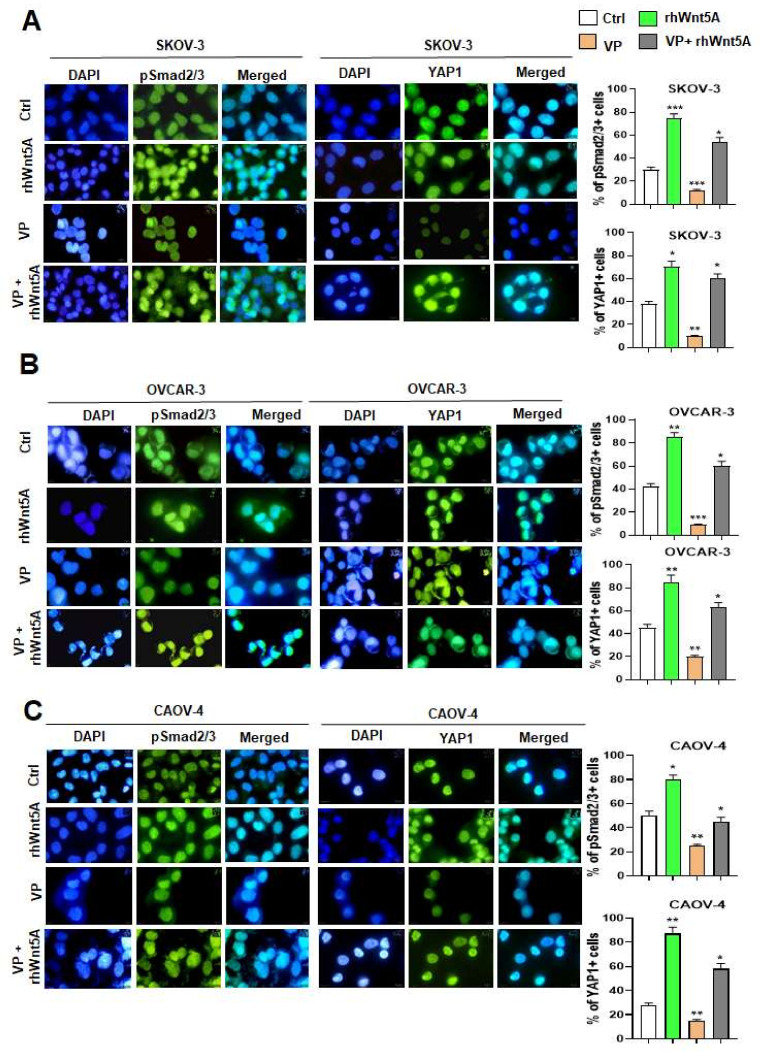
The pSmad2/3 nuclear localization is reduced in verteporfin-treated cells reverted by the presence of exogenous Wnt5A. (**A**) SKOV-3, (**B**) OVCAR-3 and (**C**) CAOV-4 Cells were treated with rhWnt5A (600 ng/mL) or Verteporfin (VP) (5 μM) alone or VP-pre-treated + rhWnt5A and subcellular localization of pSmad2/3 and YAP1. The right panels shows the percent of pSmad2/3 and YAP1 positive cells. Original magnification, ×400. *: *p* < 0.05, **: *p* < 0.01 and ***: *p* < 0.001 compared to Ctrl, n = 3. Images were visualized by Zeiss inverted fluorescence microscopy.

**Figure 6 cells-11-00237-f006:**
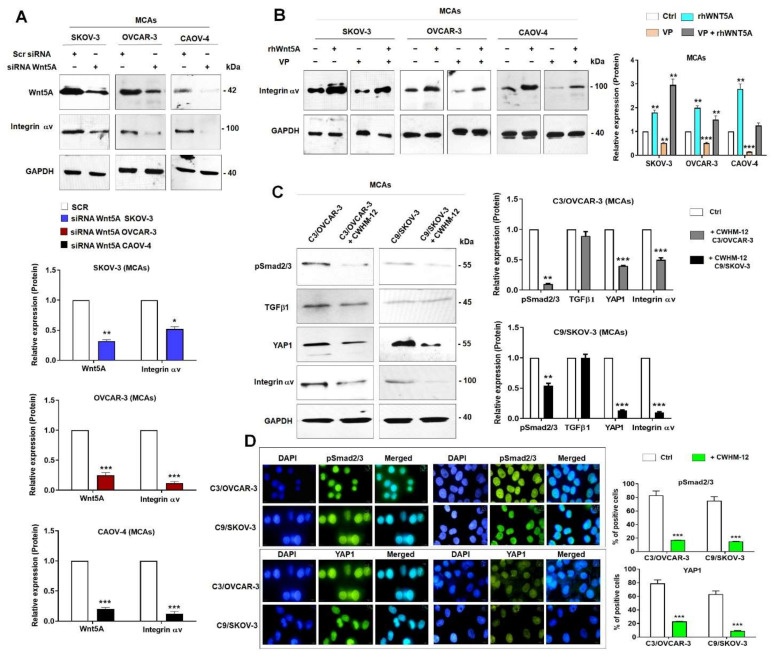
YAP1 regulates Wnt5A-induced integrin av and Smad2/3 activation. SKOV-3, OVCAR-3, and CAOV-4 cells were transfected with siRNA Scrambled (Scr) or Wnt5A. (**A**) Integrin αv expression level was assessed by immunoblotting in multicellular aggregates (MCAs) OvCa cells (upper panel) and quantifying bands from three experiments in the lower panel. (**B**) The cells were pre-treated with VP (5 μM) 1 h or treated with rhWnt5A (600 ng/mL) for 14 h alone or VP-pretreated + rhWnt5A then expression levels of integrin αv was determined in MCAs OvCa cells (Left panel), and quantification of bands (right panel). (**C**） The Wnt5A overexpressing OVCAR-3 (C3/OVCAR-3) and SKOV-3 clones (C9/SKOV-3) were treated with CWHM-12 (10 μM) for 24 h. Immunoblot of TGFβ1, pSmad2/3, YAP1 determined in MCAs OvCa cells (Left panel) and quantification of bands (right panels). The lower panel shows the quantification of bands from three independent experiments. GAPDH levels were used as an internal control, and results are expressed as mean ± SD. (D) The immunolocalization of pSmad2/3 and YAP-1 in CWHM-12-treated cells compared to control. Original magnification, ×400. *: *p* < 0.05, **: *p* < 0.01 and ***: *p* < 0.001 compared to untreated control cells (Ctrl) or Scr.

**Figure 7 cells-11-00237-f007:**
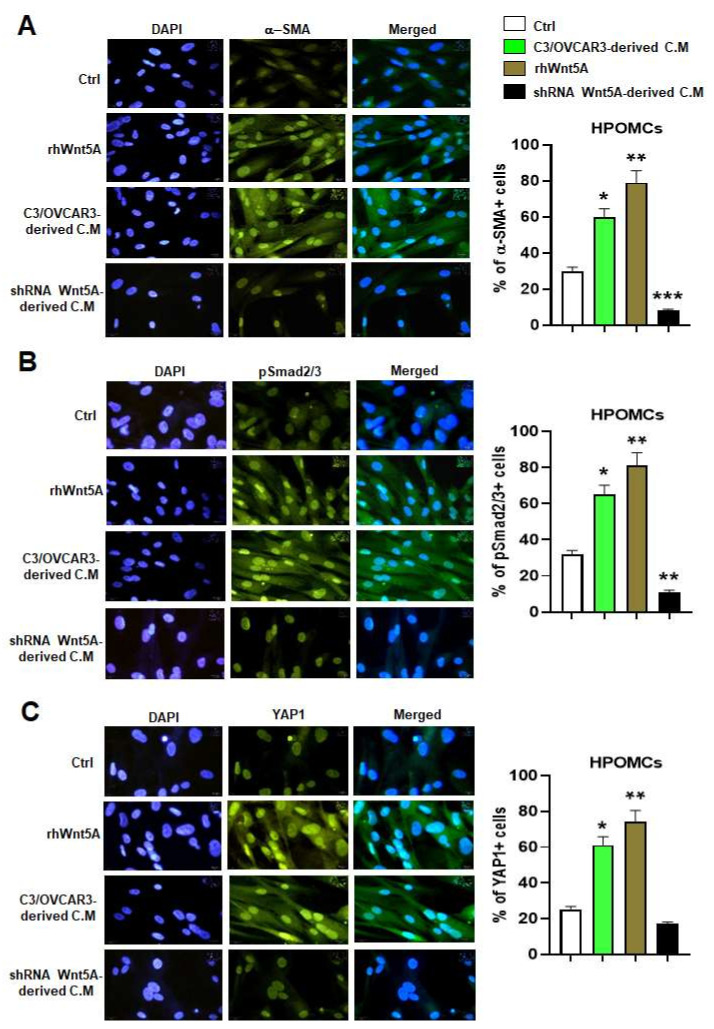
Wnt5A enhances mesothelial cell activation through Smad2/3 and YAP1 activation. The human primary omental mesothelial cells (HPOMCSs) were treated with rhWnt5A or conditioned medium (C.M) isolated from C3/OVCAR-3 clone (C3/OVCAR-3-derivedC.M), or C.M from Wnt5A silenced OVCAR-3 cells. (**A**) Immunolocalization of α-SMA, (**B**) pSmad2/3, and (**C**) YAP1. The right panels shows the percent of α-SMA, pSmad2/3, or YAP1 positive cells. Original magnification, ×400. *: *p* < 0.05, **: *p* < 0.01 and ***: *p* < 0.001 compared to Ctrl, n = 3.

**Figure 8 cells-11-00237-f008:**
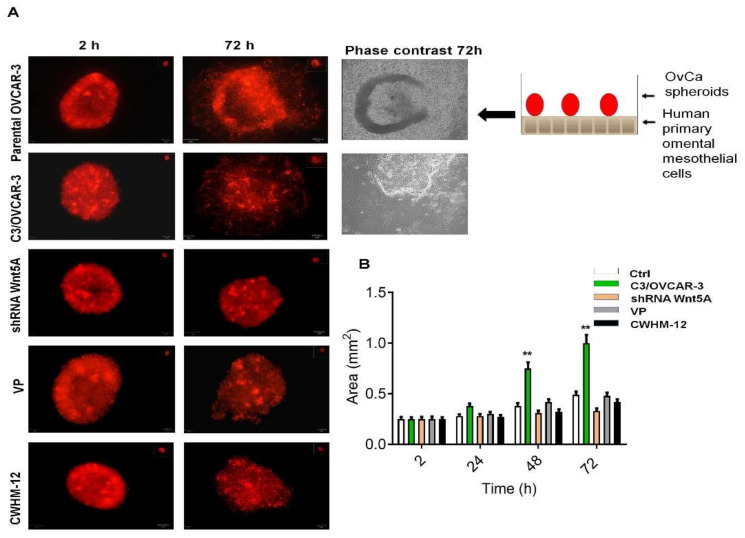
Wnt5A enhanced mesothelial cell clearance through Smad2/3 and YAP1 activation. The following OVCAR-3 spheroids were labeled with CellTracker™ CM-DiI Dye: OVCAR-3, Wnt5A overexpressing C3/OVCAR-3 spheroids, Wnt5A silenced OVCAR-3, VP-treated OVCAR-3, and CWHM-12-treated OVCAR-3 cells. Then added to the top of the HPOMCSs monolayer. (**A**) Disaggregation and invasion of multicellular aggregates (MCAs) through HPOMCSs were followed for 24, 48, and 72 h compared to the initial time set as 2 h. Phase-contrast photos after 72 h showed the mesothelial cell retraction. (**B**) The surface area of spheroids was calculated as described in material and methods. The results are expressed as mean + SD of at least three independent experiments. **: *p* < 0.01 compared to untreated OVCAR-3 spheroids (Ctrl). Original magnification: ×100.

**Figure 9 cells-11-00237-f009:**
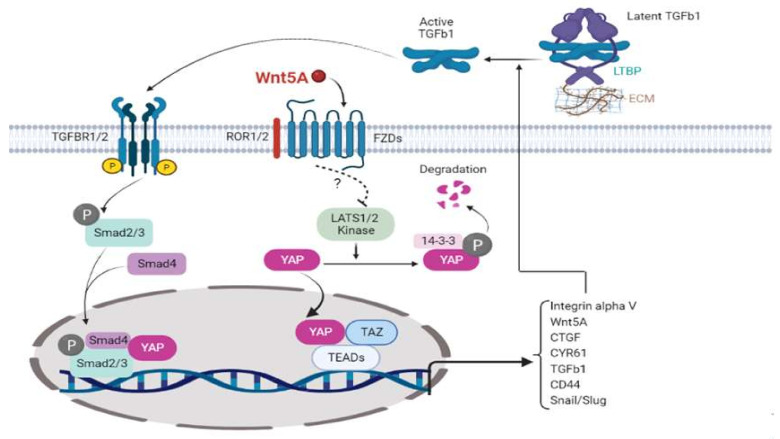
Model of Wnt5A involvement in EMT and mesothelial activation and clearance. Wnt5A derived from ovarian cancer cells through cytoskeletal rearrangement could directly or indirectly cause YAP1 phosphorylation, which translocates into the nucleus and induces TGFβ1, integrin αv, Wnt5A, and other EMT markers in this drawing. Integrin αv, in turn, may activate extracellular latent-TGFβ1 and play a pivotal role in the EMT process. In addition, YAP1 may cause retention of Smad2/3 in the nucleus, thereby prolonging their biological activity.

## Data Availability

The data presented in this study are available within the article or Appendix A.

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
