# Peer review of "Wnt5A and TGFβ1 Converges through YAP1 Activity and Integrin Alpha v Up-Regulation Promoting Epithelial to Mesenchymal Transition in Ovarian Cancer Cells and Mesothelial Cell Activation"

_cells, 2022, doi:10.3390/cells11020237_

Round 1

Reviewer 1 Report

In this article, the authors investigated the functional association between Wnt5A and the TGF-beta1/Smad2/3 and Hippo-YAP1/TAZ-TEAD pathways in ovarian cancer.  A functional role of Wnt5A in the regulation of EMT through these pathways is described. In my opinion, the manuscript lacks novelty. In fact, the role of Wnt5a in the induction of EMT is well-described in several tumor types. Moreover, the manuscript does not reach the high-quality standard required by the Journal. 

Author Response

Thank you for reviewing our work. We agree that extensive data for  TGF beta signaling involvement in EMT is not surprising but rather expected. The same applies to YAP1/Hippo pathway and integrins as executioners with strong effects on tumor cell motility and invasiveness.  Nevertheless, please note that the signaling details (which receptor, which downstream events) count here as a biological novelty. Our work has great potential, and the concept presented here should be relevant when selecting clinically beneficial targeted therapies. This implies particularly in the early stages of ovarian cancer metastasis, which needs spheroids to attach to mesothelial cells. It is well known that ovarian cancer attaches better to extracellular matrix components and fibroblasts, and in fact, attachement of ovarian cancer cells to the mesothelial layer is the rate-limiting step in ovarian cancer.

All the figures were improved and has better quality. For your consideration and review, please find below all the applied changes:

Please note that all the qRT-PCR data were transferred to the Supplementary materials file, and only western blots with 3D culture were included in all figures. Also, all figures were remodeled, and in the revised version, there are 10 instead of 13 figures. Following are the modifications:

Fig. 1 remain the same but with improved quality and enlargement

Fig. 2: includes W/B for 3D culture and improved IF images with added quantification of positive cells

Fig. 3: Improved images and histograms. qRT-PCR was transferred to supplementary material  Fig. S1A and S1B

Fig. 4: The old version of Fig. 4 and 5 were merged, represented in revision as fig. 4. All qRT-PCR data were transferred to the supplementary material file Fig. S2A and S2B

Fig. 5: IF images, improved, enlarged, and quantified. This was Fig.6 in the first submission.

Fig. 6: Improved and enlarged qRT-PCR data transferred into the supplementary material Fig. S3A. The initial submission was Fig. 7.

Fig. 7: represent the merged data of Fig. 8 and 9 in the initial submission. Also, qRT-PCR data in the old fig. 8A was transferred into the supplementary material file now numbered as Fig. S3B

Fig. 10 in the initial submission was transferred into the supplementary material file as Fig. S4.

Fig. 8: Improved, enlarged, and quantified. Initially was submitted as fig. 11

Fig. 9: Improved, enlarged, and phase contrast photos were added. Initially submitted as Fig. 12

Data for characterization and phase-contrast photos of Human primary omental mesothelial cells were transferred into Suppl. Mat file represented as Fig. S5. Initially submitted as Fig. 11A

Fig. 10: Model was re-drawn using Biorender.com. Initially submitted as Fig. 13

Reviewer 2 Report

This manuscript is expertly written and provides a deep look into the biology of a particular Wnt-pathway-related gene/protein, Wnt5a; in a defined environment - namely ovarian carcinoma cells (and tissues). 

Some of the main findings, for example, that TGF beta signaling is involved in EMT, are of course not that surprising; but rather expected. The same applies to YAP1/Hippo pathway and integrins as executioners with strong effects on tumor cell motility and invasiveness.  Nevertheless, it is the signaling details (which receptor, which downstream events) that count here as a biological novelty. 

Technical methods are described in sufficient detail (and not too much), it is good that more than 1 cell line was used for most of the core experiments. 

In the experimental part, it is to be considered as very positive that many different experiments were performed; and in most of these ( such as siRNA knockdown) a sufficient number of factors, or genes, were considered. This provides a broad enough picture of the processes analyzed and does not appear isolated or cherry-picked. 

The online data mining using "enrichr" could be explained in more detail: without actually performing an analysis yourself, it is not clear to the reader what exactly this program does, where data are derived from, which transformations are used, and - most importantly - what exactly is the purpose of the program? There are other, more transparent data mining tools out there that are easier to understand and recapitulate what actually happens. And to interpret the results. Unfortunately, figure legends don't provide any more detailed clues either. This is a critical issue and needs to be amended. 

Figure 2A: some of the Western blots appear a bit smudgy; I wonder if the authors have provided full Western blot information (= whole blots scanned) for submission, as it is now required in most journals. The densitometry measurements are too small in Fig. 2, they need to be enlarged so the reader can really see what's going on. In addition, in these densitometry plots, only a few of the genes/proteins analyzed are shown; which proteins do NOT show any significant differences in the panel on the left side? 

Similar issues apply to Fig 2B: which of these changes are statistically significant? Some of the images shown are really low resolution and should be replaced with higher resolution, higher quality pictures to drive the message home. 

Fig 3 About everything on Fig. 3 is too small and too complex, to be properly visible. Plus, image resolution is terrible, this definitely has to be fixed. Maybe split into 2 figures (3A vs. 3B). The resolution issue also applies to the invasion assays: one can get a rough hunch of what is happening, and the quantification plots on the right help... but it is not good quality. 

And again, in Fig 4, we encounter the same issues: the resolution is not sufficient, especially on the plots on the right side. There seems to be nothing wrong with the data themselves, which appear valid - but the presentation is not optimal. Also here, the complexity of the figure is high, so either enlarge the entire figure (full page?), take off some of the parts; or split. Otherwise, the quality of the Western blots appears okay; still asking for the full blots as it is standard these days. 

Yep, Fig 5 repeats the same pattern: resolution is low, complexity quite high. Nevertheless, also here - the quality of the data themselves appear fine. So this is just a technical issue, but it has to be fixed before publication. 

Fig 6: If the authors can provide a quantification of the signals shown in these microscopic images, it will make the story definitely more convincing. In some of the experiments, it's not easy to spot the differences. Numbers would help, and also provide an idea of the scope of how much is actually changing. In addition, some of the images appear out of focus and are not clear, especially with CADV4. Please look into higher quality, higher resolution, and sharp images. 

Fig 7: again, in particular, the plots on the left side are way too small. They would provide enough complexity and maybe information all by themselves. This is a pity since otherwise, the Wnt5A treatments appear to show clear effects. Also, quantification itself seems not to be the problem, it is the presentation. The results from 7A could also be shown as a heatmap? That would simplify the presentation of 3 cell lines, 3 or 4 treatments (incl. control), and 6 genes presented. Too complex, and too small...

Surprisingly, the data in Fig. 8 are much higher resolution, although also here, many different data points (all from Western blots) are crammed into very little space. It is generally the question if the authors should put all the actual plots (apart maybe from one of the figures) into supplemental data. That would clean up the high complexity of the data shown, and make it easier for the reader (and the reviewer) to appreciate what is actually shown. The biological story gets a bit blurred behind so much data, all of it shown twice: as original plots, and then again, as densitometry plots. The intention is noble, but I think it's still an overkill of information. Too much! 

Fig. 9 again has some issues with image resolution, although the WBs appear okay - the images of the cells are not. 

Generally, there is A LOT of data and MANY figures. I think the volume should be somewhat reduced. Too much data, too much background, and details are not helping the story. Maybe try to focus, present only the most relevant stuff. This is very intense and the reader gets a bit tired by Fig. 9 to go through the many results. The narrative, as such, is still fine and the biological message is also clear - but it's a lot of clutter. 

There is a reason why most journals have a word and a page limit, and also limit the number of figures allowed (usually between 4 and 6 or so). This should also be applied here! There is no problem removing some of the material to the supplemental files and showing it there. But even if that is done, I think the supplemental file could be reduced to a reasonable size (no more than 10 extra figures). 

Fig. 10: finally, this on is not too complex and the resolution appears okay. Message is also clear. 

Fig 11: also this one is just fine; the images of cells are of much higher resolution and better quality than in any of the other figures. So it is possible to provide higher quality image materials. Please, consider fixing this in the other figures as well. Or, wherever this is not possible, remove those figures, and place them in supplements. 

Fig 12: I don't know what microscope has been used - but the images of these organoids are NOT high enough quality and resolution.

Discussion is fine, and not too long; but as with everything in this manuscript - its getting a bit lengthy in size. Also here, focusing on the most relevant issues would be beneficial. 

In short: the biological message of this manuscript is fine, but the presentation definitely needs more focus - and better quality image/figure material. 

Author Response

This manuscript is expertly written and provides a deep look into the biology of a particular Wnt-pathway-related gene/protein, Wnt5a, in a defined environment - namely ovarian carcinoma cells (and tissues). 

Some of the main findings, for example, that TGF beta signaling is involved in EMT, are of course not that surprising; but rather expected. The same applies to YAP1/Hippo pathway and integrins as executioners with strong effects on tumor cell motility and invasiveness.  Nevertheless, it is the signaling details (which receptor, which downstream events) that count here as a biological novelty. 

Technical methods are described in sufficient detail (and not too much); it is good that more than 1 cell line was used for most of the core experiments. 

In the experimental part, it is to be considered as very positive that many different experiments were performed; and in most of these ( such as siRNA knockdown), a sufficient number of factors, or genes, were considered. This provides a broad enough picture of the processes analyzed and does not appear isolated or cherry-picked. 

The online data mining using "enrich" could be explained in more detail: without actually performing an analysis yourself, it is not clear to the reader what exactly this program does, where data are derived from, which transformations are used, and - most importantly - what exactly is the purpose of the program? There are other, more transparent data mining tools out there that are easier to understand and recapitulate what happens. And to interpret the results. Unfortunately, figure legends don't provide any more detailed clues either. This is a critical issue and needs to be amended. 

Author reply: Thank you very much for your thoughtful and constructive comments. Your point of view certainly reinforces and encourages us to continue our rational thinking and research. Regarding enrichr we did not perform any transformation of data. We applied enrichment analysis of TFs (transcription factors) and GO (gene ontology) in upregulated assessed genes here, including the TGFb family, Wnt5A, and YAP/TAZ components in human ovarian cancer samples. Also, Kyoto Encyclopedia of Genes and Genomes (KEGG) analyses were performed for TGFb and Wnt5A pathways using Enrichr (http://amp.pharm.mssm.edu/Enrichr). This has been added and highlighted on page 5, lines 175-181.

Figure 2A: some of the Western blots appear a bit smudgy; I wonder if the authors have provided full Western blot information (= whole blots scanned) for submission, as it is now required in most journals. The densitometry measurements are too small in Fig. 2, they need to be enlarged so the reader can see what's going on. In addition, in these densitometry plots, only a few of the genes/proteins analyzed are shown; which proteins do NOT show any significant differences in the panel on the left side? Similar issues apply to Fig 2B: which of these statistically significant changes? Some of the images shown are really low resolution and should be replaced with higher resolution, higher quality pictures to drive the message home. IF image should be improved and quantified

Author reply: Thank you very much for your time, constructive comments, and careful review of our work. We want to take the opportunity and emphasize the fact that the early events of ovarian cancer metastasis require the formation of ovarian cancer cells aggregates or spheroids, preventing anoikis. Thus, MCAs could better represent the biological response of cells to different culture conditions.

Therefore, our three-dimensional culture in our work can be more representative of physiological conditions. In addition, the results of monolayer culture conditions did not show a significant difference with data obtained with MCAs (Fig. 2A). Hence, for ease of understanding and following the content, we modified Fig. 2, including the 3D culture and IF image. Fig 2B was improved, and in the revised version, the percent of pSmad2/3 positive cells for each treatment condition has been added and shown in histogram (left panel). All the figures were enlarged, and quantification of all genes was included.

Fig 3 About everything on Fig. 3 is too small and too complex, to be properly visible. Plus, image resolution is terrible, this definitely has to be fixed. Maybe split into 2 figures (3A vs. 3B). The resolution issue also applies to the invasion assays: one can get a rough hunch of what is happening, and the quantification plots on the right help... but it is not good quality. 

Author reply: Thank you we considered your suggestion. Figure 3 was improved and enlarged, and all qPCR data were transferred to the Supplementary material file (Fig. S1A and S1B) for the ease of understanding and following the main text and figures.

Please note that this applies to every initially submitted figure with qRT-PCR data. In the revised version, all the qRT-PCR data were transferred to the Supplementary materials file, and for all the figures, only western blots with 3D culture were included since there is no significant difference between 2D and 3D, and 3D results are more representative of physiological condition.

And again, in Fig 4, we encounter the same issues: the resolution is not sufficient, especially on the plots on the right side. There seems to be nothing wrong with the data themselves, which appear valid - but the presentation is not optimal. Also here, the complexity of the figure is high, so either enlarge the entire figure (full page?), take off some of the parts; or split. Otherwise, the quality of the Western blots appears okay; still asking for the full blots as it is standard these days. 

Author reply: Thank you we considered your suggestion. Figure 4 was modified, improved, and enlarged. and all qPCR data were transferred to the Supplementary material file (Fig. S2A and S2B)

Yep, Fig 5 repeats the same pattern: resolution is low, complexity quite high. Nevertheless, also here - the quality of the data themselves appear fine. So this is just a technical issue, but it has to be fixed before publication. 

Author reply: Thank you for your valuable comment. Please note that the old figure 5  is now part of figure 4 (B panel) was improved and enlarged.

Fig 6: If the authors can provide a quantification of the signals shown in these microscopic images, it will make the story definitely more convincing. In some of the experiments, it's not easy to spot the differences. Numbers would help, and also provide an idea of the scope of how much is actually changing. In addition, some of the images appear out of focus and are not clear, especially with CADV4. Please look into higher quality, higher resolution, and sharp images.quantification is needed 

Author reply: Thank you for your helpful suggestion. The old figure 6 is in the revised version shown as Fig. 5, which has been improved, enlarged, and quantified.

Fig 7: again, in particular, the plots on the left side are way too small. They would provide enough complexity and maybe information all by themselves. This is a pity since otherwise, the Wnt5A treatments appear to show clear effects. Also, quantification itself seems not to be the problem, it is the presentation. The results from 7A could also be shown as a heatmap? That would simplify the presentation of 3 cell lines, 3 or 4 treatments (incl. control), and 6 genes presented. Too complex, and too small

Author reply: Thank you we considered your suggestion and did our best. Please note that in the revised version this figure is numbered as fig. 6 which was improved and enlarged. All qRT-PCR data were transferred to the Supplementary file  Fig. S3A.  

 Surprisingly, the data in Fig. 8 are much higher resolution, although also here, many different data points (all from Western blots) are crammed into very little space. It is generally the question if the authors should put all the actual plots (apart maybe from one of the figures) into supplemental data. That would clean up the high complexity of the data shown, and make it easier for the reader (and the reviewer) to appreciate what is actually shown. The biological story gets a bit blurred behind so much data, all of it shown twice: as original plots, and then again, as densitometry plots. The intention is noble, but I think it's still an overkill of information. Too much! 

Author reply: Thank you for your suggestion. Here, the initially submitted fig. 8 is now fig. 7. This figure may contain a lot of data; we tried our best to keep it simple! (we have no choice; there is much data, and we wanted to reduce the number of figures from 13 to 10 in the revised version). All qRT-PCR data were transferred to the supplementary file Fig. S3B, and all images were improved and enlarged for better visibility.

Fig. 9 again has some issues with image resolution, although the WBs appears okay - the images of the cells are not. Quantification

Author reply: Thank you for your suggestion. Now the initially submitted fig. 9 became part of fig.7 (7C and panels) and IF images were improved and quantified.

Generally, there is A LOT of data and MANY figures. I think the volume should be somewhat reduced. Too much data, too much background, and details are not helping the story. Maybe try to focus, present only the most relevant stuff. This is very intense and the reader gets a bit tired by Fig. 9 to go through the many results. The narrative, as such, is still fine and the biological message is also clear - but it's a lot of clutter. 

There is a reason why most journals have a word and a page limit, and also limit the number of figures allowed (usually between 4 and 6 or so). This should also be applied here! There is no problem removing some of the material to the supplemental files and showing it there. But even if that is done, I think the supplemental file could be reduced to a reasonable size (no more than 10 extra figures). 

Author reply: Thank you; we are aware of this issue and truly apologize for the difficult reading of the first version. All your suggestions were taken carefully into consideration, and the number of figures was reduced to 10. So three figures less. All qRT-PCR data were transferred into the Suppl material file. We hope these modifications make the reading of this manuscript easier and we hope more pleasant.

improve 10: finally, this on is not too complex and the resolution appears okay. Message is also clear. 

Author reply: Thank you. The initial submitted Figure 10 is now numbered as supplementary figure S4, improved and enlarged.

Fig 11: also, this one is just fine; the images of cells are of much higher resolution and better quality than in any of the other figures. So it is possible to provide higher quality image materials. Please, consider fixing this in the other figures as well. Or, wherever this is not possible, remove those figures, and place them in supplements. 

Author reply: Thank you we considered your suggestion. The initially submitted figure 11 now is numbered as fig. 8 improved, enlarged, and quantified. Also initially submitted Fig. 11A now is transferred into the supplementary material file as Fig. S5.

Fig 12: I don't know what microscope has been used - but the images of these organoids are NOT high enough quality and resolution.

Author reply: Thank you for your insightful comment. Our institution has no confocal microscopy and for us taking this quality of photos is a challenge using an inverted fluorescence Zeiss microscope. We think the quality of the photos is acceptable. We selected the best photos to improve, enlarge, and quantify the images. Please note that the initially submitted Fig. 12 is now numbered as Fig. 9.

Discussion is fine, and not too long; but as with everything in this manuscript - its getting a bit lengthy in size. Also here, focusing on the most relevant issues would be beneficial. 

Author reply: We took into consideration although we have to discuss the crosstalk between different signaling pathways, we tried our best and shortened the discussion.

In short: the biological message of this manuscript is fine, but the presentation definitely needs more focus - and better quality image/figure material. 

Author reply: All the figures were improved, our model as fig. 10 was drawn using Biorender.com in this version, and we did our best to present them more clearly. 

For the good sake, all changes were summarized below:

Please note that all the qRT-PCR data were transferred to the Supplementary materials file, and only western blots with 3D culture were included in all figures. Also, all figures were remodeled, and in the revised version, there are 10 instead of 13 figures. Following are the modifications:

Fig. 1 remain the same but with improved quality and enlargement

Fig. 2: includes W/B for 3D culture and improved IF images with added quantification of positive cells

Fig. 3: Improved images and histograms. qRT-PCR was transferred to supplementary material  Fig. S1A and S1B

Fig. 4: The old version of Fig. 4 and 5 were merged, represented in revision as fig. 4. All qRT-PCR data were transferred to the supplementary material file Fig. S2A and S2B

Fig. 5: IF images, improved, enlarged, and quantified. This was Fig.6 in the first submission.

Fig. 6: Improved and enlarged qRT-PCR data transferred into the supplementary material Fig. S3A. The initial submission was Fig. 7.

Fig. 7: represent the merged data of Fig. 8 and 9 in the initial submission. Also, qRT-PCR data in the old fig. 8A was transferred into the supplementary material file now numbered as Fig. S3B

Fig. 10 in the initial submission was transferred into the supplementary material file as Fig. S4.

Fig. 8: Improved, enlarged, and quantified. Initially was submitted as fig. 11

Fig. 9: Improved, enlarged, and phase contrast photos were added. Initially submitted as Fig. 12

Data for characterization and phase-contrast photos of Human primary omental mesothelial cells were transferred into Suppl. Mat file represented as Fig. S5. Initially submitted as Fig. 11A

Fig. 10: Model was re-drawn using Biorender.com. Initially submitted as Fig. 13

Reviewer 3 Report

The manuscript entitled " Wnt5A and TGFB1 Converges through YAP1 Activity and Integrin alpha v Up-regulation Promoting Epithelial to Mesenchymal Transition in Ovarian Cancer Cells and Mesothelial Cell Activation" by Zeinab et al describes Wnt5A was implicated in the activation of human omentum derived mesothelial cells subsequent invasion of ovarian cancer cells and indicate Wnt5A could be a critical mediator of EMT-associated pathways. The paper has great potential and indeed there are exciting implications for ovarian cancer therapeutics. Nonetheless, there were some concerns/suggestions that the authors should address.

  1. Fig2A, the efficiency of Wnt5A knockdown is not enough.at least 50% knockdown can be used for further steps. For quantification parts, ML and MCAs are different conditions, you should not use one control for two different experiments.
  2. For Result 3, you should give full name about CDH2, VIM, FN and CD44 since it is the first time to be seen in the paper and explain why you choose these four for this paper and it looks random for me.
  3. For result 3,since protein is the embodiment of the main function, you should check CDH2, VIM, FN and CD44 protein level after different treatments. You also need check other EMT makers also.
  4. Fig 2A, Fig 4A and Fig8A, Wnt5A siRNA blots, you used same blots for all three experiments, please double check it.

For all legend part, you should follow the magazine’s requirements for a unified format.

Round 2

Reviewer 1 Report

The paper has been improved. I have only some minor comments:

  • Fig. 7C. MW markers should be aligned.
  • Fig. 7D. In the histograms, statistical analysis is missing.

Author Response

Reviewer 1: The paper has been improved. I have only some minor comments:

  • 7C. MW markers should be aligned.
  • Author reply: Thank you for the careful review of our manuscript. Please note that, as one of the reviewers suggested, we reduced the number of figures; thus, we transferred fig. 6 to the suppl file. Hence Initially submitted Fig. 7 now is Fig. 6 in which MWs were aligned.
  • 7D. In the histograms, statistical analysis is missing.
  • Author reply: Thank you for the detailed review. Statistical analysis is inserted.

Reviewer 2 Report

The authors have taken the reviewer's comments all very seriously. and the manuscript has very significantly improved. Instead of 13 complex figures, there are now only 10 of reduced complexity. Well, that being said, I still think 10 figures is a lot, and the complexity of some of them is still very high. Specifically, I think figures 5 and 8, showing many microscopic images of cells, are very complex and still a bit confusing. But it's getting there. Further reducing the density in these figures may only be beneficial There may also be issues with the font sizes used within the figures. I don't know if there are recommendations or limits given by the journal, but maybe 1) they should be generally a bit larger and 2) they should also be more homogeneous... wherever possible, use the same or similar fonts, and font sizes, across the entire figure. It helps the reader to appreciate the content of the figure. This becomes evident, for example, in Figure 5, where the font sizes describing the bar graphs are really small. 

Author Response

Reviewer 2: The authors have taken the reviewer's comments all very seriously. and the manuscript has very significantly improved. Instead of 13 complex figures, there are now only 10 of reduced complexity. Well, that being said, I still think 10 figures is a lot, and the complexity of some of them is still very high.

Author reply: Thank you for your insightful comments, which helped us improve our manuscript. Fig. 6 in the main text was transferred to the supplementary material file named Fig. S3 for reducing the number of figures in the main text. Hence the paper contains the total of 9 Figures in the second revision word template.

Specifically, I think figures 5 and 8, showing many microscopic images of cells, are very complex and still a bit confusing. But it's getting there. Further reducing the density in these figures may only be beneficial There may also be issues with the font sizes used within the figures. I don't know if there are recommendations or limits given by the journal, but maybe 1) they should be generally a bit larger and 2) they should also be more homogeneous... wherever possible, use the same or similar fonts, and font sizes, across the entire figure. It helps the reader to appreciate the content of the figure. This becomes evident, for example, in Figure 5, where the font sizes describing the bar graphs are really small

Author reply: The arrangement of photos was changed, and it seems now it is easier to follow and understand. The changes in Fig. 5 consist of the separation of cell lines for immunostaining with different antibodies. Thus SKOV-3 photos are in panel 5A, OVCAR-3 photos in panel B, and CAOV-4 photos in panel C. Quantification for each cell line is represented beside (right) of each panel.

The submitted fig. 8 in the revised file is now numbered as Fig. 7 and rearranged so that each antibody staining is represented in different panels. Also, due to the lack of place, the former fig. 8C now is in the supplementary file as Fig. S6C.

We did our best to unify the font size.

To summarize the changes in the second revision:

Fig. 6 was transferred to suppl. The file named Fig. S3

Initially submitted fig. 8 now is Fig. 7

Initially submitted Fig. 9, now is Fig. 8

Initially submitted Fig. 10, now is Fig. 9

The supplementary file contains 6 figures in total

Reviewer 3 Report

The paper has been improved and meets the published quality.

Author Response

Reviewer 3: The paper has been improved and meets the published quality.

Author reply: Thank you for your consideration and time to review our work to improve it.